# Association between Healthy Eating Index and Mental Health in Middle-Aged Adults Based on Household Size in Korea

**DOI:** 10.3390/ijerph19084692

**Published:** 2022-04-13

**Authors:** Ji-Myung Kim, EunJung Lee

**Affiliations:** 1Food and Nutrition Major, Division of Food Science & Culinary Arts, Shinhan University, Uijeongbu 11644, Gyeonggi-do, Korea; kjm@shinhan.ac.kr; 2Food and Nutrition Major, School of Wellness Industry Convergence, Hankyong National University, Jungang-no, Anseong-si 17579, Gyeonggi-do, Korea

**Keywords:** Healthy Eating Index, mental health, single-person, middle-aged, Korean

## Abstract

This study examined the association between dietary lifestyle and mental health outcomes among middle-aged individuals (40–60-year-olds) living alone, using the Korean Healthy Eating Index (KHEI). The participants were selected (1334 men and 2185 women) from the Korea National Health and Nutrition Examination Survey 2016–2018 and categorized into single/multi-person households. The KHEI scores were calculated based on the 24 h recall data of dietary intake. Among both sexes, single-person households had significantly lower subjective health status scores (*p* = 0.015 for men; *p* < 0.001 for women), lower EuroQol-5D index scores (*p* = 0.011 for men; *p* = 0.003 for women), and higher Patient Health Questionnaire-9 index scores (*p* = 0.004 for men; *p* < 0.001 for women), compared to multi-person households. Men had a higher risk of depression in single-person households compared to multi-person households (OR, 3.5; 95% CI, 1.2–10.1). For women, the ORs for stress perception with the first tertile group of KHEI scores in single-person and multi-person households compared to the third tertile group were 3.5 (95% CI, 1.3–9.0) and 1.4 (95% CI, 1.0–1.8), respectively. The results of this study are expected to be used as baseline data to establish nutrition, healthy eating behavior, and health policies for the middle-aged living alone.

## 1. Introduction

The indices used to evaluate the quality of meals in adults and the elderly include the American Dietary Quality Index (DQI) [1], the Healthy Eating Index (HEI) [2], and the Mediterranean diet score (MDS) [3]. The Korean Healthy Eating Index (KHEI) considers the characteristics of Korean meals and national dietary guidelines. It has been developed to monitor and evaluate the eating habits of a population group through data such as the National Health and Nutrition Survey, rather than assessing individual eating habits [4]. People’s eating habits have become very diverse, and it is necessary to evaluate the overall quality of eating habits using these dietary evaluation indexes. Especially, meals in single-person households often lack variety and specific core foods, such as fruits, vegetables, and fish, and are associated with unhealthy dietary patterns [5].

The proportion of single-person households worldwide is increasing due to various reasons, such as an increased rate of late marriage and divorce and young adults choosing to live independently. South Korea has registered a three-fold growth over the last 30 years (from 9.0% in 1990 to 27.1% in 2015) and is expected to account for more than one-third of all single households by 2035 [6,7]. The increase in middle-aged single-person households is attributed to singles, divorcees, or goose fathers, who live and work alone away from their families to pay for living and educational costs for their families [8,9,10]. Single-person households are more exposed to risks such as income, housing, safety, and health compared to multi-person households. In the case of single-person households, due to social isolation, depression, loneliness, and sadness are greater than those of multi-person households, and they are not active in participating in cultural activities [11]. In single-person households, health care utilization such as physician visits and hospitalization are substantially higher, and the proportions of chronic disease prevalence such as gastritis and mental disorders are also greater than in non-single households [12,13]. Henning-Smith and Gonzales suggested that middle-aged adults living alone suffer poor self-rated health compared with those not living alone [14].

A related concept to subjective health is quality of life. The WHO defines the quality of life as “individual’s perception of their position in life within the context of the culture and value systems in which they live and in relation to their goals, expectations, standards and concerns” [15]. Living alone is known to be associated with a number of adversities such as poor mental and physical health, loneliness, and mortality [16,17]. Health stress and economic stress have been identified as factors that increase depression in single-person households [18]. The rates of suspected depression (27.2 vs. 8.8%) and suicidal ideation (13.9 vs. 3.0%) indicate that the mental health problem of middle-aged adults in single households could be three times more than in non-single households [12]. A systematic review by Tamminen et al. [19] reported that a potential association between living alone and low positive mental health was found in three out of the four studies. Otherwise, according to previous studies related to diet and mental health, many studies have been conducted on food and nutrition intake status and diet quality among adolescents [20], adults [21], and the elderly [22,23]. A systematic review by Głąbska et al. [24] reported that a high total intake of fruits and vegetables might protect against depressive symptoms. In addition, many studies have reported the relationship between dietary evaluation indices such as HEI [25,26,27] and alternative HEI [28] and mental health. Recently, Yoon and Oh [29] reported the relationship between KHEI and psychological distress in Korean adults. However, there are few studies on the relationship between KHEI and mental health.

This study aimed to identify the association between Korean Healthy Eating Index (KHEI) scores and mental health in middle-aged adults living in single-person households, based on the 2016–2018 Korea National Health and Nutrition Examination Survey (KNHANES VII-1). The overall goal of the study is to guide those with stress and depression toward healthy eating behaviors by providing information on basic dietary data.

## 2. Materials and Methods

### 2.1. Study Design and Participants

The data used in this study were extracted from KNHANES 2016–2018, conducted by the Korea Centers for Disease Control and Prevention (KCDC). KNHANES is a complex, stratified, multistage probability, and cluster survey conducted across the year in a rolling method to sample participants representing the Korean population. KNHANES consists of health interviews, examinations, and nutrition surveys in a cross-sectional design. Details regarding KNHANES are described in the study by Kweon et al. [30]. The survey protocol was approved by the Institutional Review Board of the KCDC (approval no. 2018-01-03-P-A), and all participants provided informed consent.

Participant selection was based on several factors. Among the 6198 who were aged 40–60 who participated in all three surveys (health and behavior interviews; health examinations; nutrition surveys), participants with abnormal dietary intake (men with <800 kcal/day or >4000 kcal/day and women with <500 kcal/day or >3500 kcal/day) (*n* = 278) were excluded [31]. Participants whose data did not include the variables of subjective health status, stress, depression, and quality of life (*n* = 2399), and those with missing socio-demographic variables (*n* = 3) were also excluded. Subsequently, 3519 participants (1334 men and 2185 women) were included in this study.

### 2.2. General Measurements

Socio-demographic factors, such as age, sex, education, household income, marital status, and household size, were collected using a self-reported questionnaire. Smoking status, alcohol consumption, and physical activity data were collected using health and behavior interviews. The levels of education were categorized in accordance with their graduation status (elementary, middle, high school, or higher). Marital status was categorized as either married or single and household income as low, middle-low, middle-high, and high. Smoking status was categorized as current or past/never and alcohol intake as non-drinker or drinker. The household size was categorized as single-person or multi-person.

Anthropometric data were collected during the health examinations. Body mass index (BMI) was calculated as weight (kg) divided by height (m) squared (kg/m^2^) and the participants were classified into four categories based on their BMIs (kg/m^2^): underweight (BMI < 18.5), normal weight (18.5 ≤ BMI < 23), overweight (23 ≤ BMI < 25), or obesity (BMI ≥ 25) [32].

### 2.3. Mental Health Behavior Measures

#### 2.3.1. Subjective Health Status

Subjective health status data were measured using one item in a health and behavior interview. Participants were asked, “In general, how would you rate your health?” The participants answered on a five-point scale of 1 (very bad), 2 (bad), 3 (moderate), 4 (good), or 5 (very good). In this study, subjective health status was classified into two categories: good (self-rating of very good, good, or moderate) or poor health (self-rating of bad or very bad) [33].

#### 2.3.2. Stress Perception

Stress perception was measured with one item to check the level of stress experienced in daily life. The participants answered on a four-point scale of 1 (hardly stressed), 2 (little stressed), 3 (quite stressed), or 4 (extremely stressed). When the participants responded to feeling “extremely or quite stressed” in their daily lives, the stress level was classified as “yes.” The “a little or hardly stressed” was classified as no stress perception [34].

#### 2.3.3. Depression

Participants with depression were identified based on their responses to the Patient Health Questionnaire (PHQ)-9 section of the health survey. The participants were asked, “How often have you been bothered by any of the following symptoms over the last two weeks?” The PHQ-9 uses nine items to measure the severity of depressive symptoms. Each of the nine items was rated on a four-point scale of 0 (not at all), 1 (several days), 2 (more than half the days), and 3 (every day), and the answers were summed to obtain the total PHQ-9 score. Based on the methods used in previous studies, in this study participants with a PHQ-9 score ≥ 10 (27 points) were defined as having depression [35].

#### 2.3.4. Health-Related Quality of Life

Health-related quality of life was measured by EuroQol-5D (EQ-5D) questionnaire of the health survey, which was developed by the EuroQoL Group. EQ-5D was descriptively quantified by five dimensions, namely mobility, self-care, usual activities, pain/discomfort, and anxiety/depression. Each dimension was described as follows: 1 = “no problem”, 2 = “some problems”, and 3 = “severe problems”. All of these response levels were converted into an EQ-5D index using the weight scoring system of the five dimensions ranging from 0 (worst) to 1 (best) [36].

### 2.4. Dietary Intake Measures

#### 2.4.1. Dietary Intake

Dietary intake data were obtained using a 24 h dietary recall protocol as part of the dietary survey. The data were collected by trained dietitians at participants’ homes a week after the health interview and health examination surveys were completed. However, there are some disadvantages of this method; it relies on the participant’s memory and is not able to describe the typical diet with a single day’s intake. Using a 24 h dietary recall, total energy and nutrient intake were calculated using a food composition table published by the Rural Development Administration of Korea [4]. Total daily energy intake, percentage of energy intake from macronutrients (carbohydrate, protein, fat), and nutrient amount per 1000 kcal were evaluated.

#### 2.4.2. KHEI

KHEI is an indicator developed by KCDC as a means of assessing adherence to national dietary guidelines and comprehensive dietary life and quality among Koreans [35]. It has 14 components overall: eight evaluating the adequacy of the recommended food and nutrient intake (having breakfast, multi-grains, fruits, fresh fruits, vegetables, vegetables excluding kimchi and pickles, milk and dairy products, meat, fish, eggs, legumes); three evaluating the intake of food and nutrients for which consumption restriction is recommended (percentage of energy intake from saturated fatty acids, sweets and beverages, and sodium); and three evaluating the balance of energy intake (percentage of energy intake from carbohydrates, fats, adequate energy intake) (Appendix A). Sweets and beverages included sugars, confectionary, coffee or tea, cocoa, alcoholic beverages, soft drinks, fruit and vegetable drinks, and other beverages. The maximum possible score of the KHEI is 100 points. Some of the components weigh 5 points (fruits, vegetables, multi-grains, percentage of energy intake from carbohydrates and fats, adequate energy intake), and the rest weigh 10 points given their importance. The KHEI score for each component is outlined in the KNHANES raw data.

### 2.5. Statistical Analysis

All the statistical analyses were performed using SAS software (version 9.4; SAS Institute, Cary, NC, USA). To prevent biased results, the complex survey design consisted of multistage, stratified, and clustered samples and survey weights that reflected the estimates of the entire Korean population. The continuous variables, including age, BMI, subjective health status, stress perception, PHQ-9 scores, EQ-5D index, intake of the total energy, nutrient intake, and KHEI scores, were expressed as means ± standard errors. The categorical variables, including age, education, household income, marital status, alcohol intake, smoking, physical activity, BMI, subjective health status, stress, and depression categories, were expressed as frequencies and percentages.

The differences by sex and household size were compared using the Chi-square test and a general linear model. To analyze the association between KHEI scores by sex and household size (tertile) and mental health outcomes (unhealthy status, stress, and depression), the odds ratios (ORs) and the 95% confidence intervals (CIs) were calculated using logistic regression analyses. The prevalence of unhealthy status, stress level, and depression were compared by sex and household size, using multiple logistic regression analyses before (Model 1) and after adjusting for variables (Model 2). The adjustment variables were age, marital status, household income levels, education levels, and smoking status in men, and age, education levels, marital status, household income levels, and smoking status in women. The statistical significance was set at *p* < 0.05.

## 3. Results

### 3.1. General Characteristics of Participants

Table 1 shows the demographic characteristics of the participants according to household size by sex. The mean age of male participants was 49.19 ± 0.54 years in single-person households and 49.59 ± 0.18 years in multi-person households. The mean age of female participants was 51.11 ± 0.61 years in single-person households and 49.65 ± 0.16 years in multi-person households (*p* = 0.021). Among both sexes, single-person households were significantly more likely to have low household income (*p* = 0.001 for men; *p* < 0.0001 for women), be single (*p* < 0.0001 for men and women), and be current smokers (*p* < 0.0001 for men and women) than multi-person households. Among women, single-person households were significantly more likely to have high school degrees or higher than multi-person households (*p* < 0.001).

### 3.2. Mental Health-Related Outcomes of Participants

Table 2 shows participants’ mental health-related outcomes such as subjective health status, stress perception, PHQ-9 index, and EQ-5D index according to household size by sex. In both sexes, single-person households showed significantly lower subjective health status scores (*p* = 0.015 for men; *p* < 0.001 for women), lower EQ-5D index scores (*p* = 0.011 for men; *p* = 0.003 for women), and higher PHQ-9 index scores (*p* = 0.004 for men; *p* < 0.001 for women) compared to multi-person households. Among both sexes, the rates of poor health (24.98 vs. 13.05%, *p* < 0.001 for men; 30.05 vs. 16.82%, *p* < 0.001 for women) and depression (12.59 vs. 2.25%, *p* < 0.0001 for men; 10.56 vs. 3.65%, *p* < 0.0001 for women) were significantly higher in single-person households than in multi-person households.

### 3.3. Daily Nutrient Intake of Participants

Table 3 shows participants’ daily nutrient intakes. In both sexes, there was no significant difference in total energy intake by household size. In addition, the proportions of carbohydrate, protein, and fat intake did not significantly differ. However, among women, protein density (*p* = 0.031) in single-person households was significantly lower and vitamin A density (*p* = 0.042) was significantly higher than that of multi-person households.

### 3.4. KHEI of Participants

Table 4 shows the participants’ KHEI scores. Among men, single-person households had a significantly lower overall KHEI score compared to multi-person households (*p* < 0.0001). Moreover, in men, single-person households had a lower KHEI component score for breakfast (*p* < 0.0001), mixed grains (*p* < 0.0001), total fruit (*p* = 0.003), fresh fruit (*p* < 0.0001) intake, and a moderate intake of sweets and beverages (*p* = 0.049) compared to the multi-person households. However, among women, there was no significant difference in the total KHEI score and component scores of KHEI by household size.

### 3.5. Relationship between Household Size and Mental Health-Related Outcomes

Table 5 presents the ORs for mental health-related outcomes according to the household size by sex. In both sexes, single-person households were associated significantly with a higher prevalence of poor health (Model 1: OR, 2.2; 95% CI, 1.4–3.6, *p* = 0.001 for men; OR, 2.1; 95% CI, 1.4–3.2, *p* < 0.001 for women) and depression (Model 1: OR, 6.3; 95% CI, 2.7–14.3, *p* < 0.0001 for men; OR, 3.1; 95% CI, 1.7–5.6, *p* < 0.001 for women) compared with multi-person households. After adjustment for confounding factors, in only men, single-person households were associated significantly with a higher prevalence of depression compared with multi-person households (Model 2: OR, 3.5; 95% CI, 1.2–10.1, *p* = 0.018).

### 3.6. Relationship between KHEI and Mental Health-Related Outcomes

Table 6 presents the ORs for mental health-related outcomes according to the tertiles of the KHEI scores by household size in men. For men, there was no significant association with the prevalence of poor health, stress perception, and depression according to the tertiles of the KHEI scores by household size in Models 1 and 2.

Table 7 presents the ORs for mental health-related outcomes according to the tertiles of the KHEI scores by household size in women. For women, the ORs for stress perception in the first tertile (T1) group in single-person and multi-person households compared to the third tertile (T3) group were 3.5 (95% CI, 1.3–9.0) and 1.4 (95% CI, 1.0–1.8), respectively. The ORs for depression in the T1 group in single-person households and multi-person households compared to the T3 group were 25.3 (95% CI, 3.2–203.8) and 2.2 (95% CI, 1.1–4.6), respectively, and significant dose–response associations were observed (*p* for trend < 0.05). After adjustment for confounding factors, the adjusted ORs for stress perception and depression with the T1 group compared to the T3 group in only single-person households were 2.8 (95% CI, 1.0–7.5) and 28.3 (95% CI, 2.1–374.2), respectively, and significant dose–response association was observed only for depression (*p* for trend = 0.007).

In addition, in all subjects, the adjusted OR for depression with the T1 group compared to the T3 group was 2.2 (95% CI, 1.3–4.0), and significant dose–response association was observed for depression (*p* for trend = 0.010) (data not shown).

## 4. Discussion

This study was designed to identify the association between healthy eating index and mental health among middle-aged adults in single-person households in Korea based on the 2016–2018 KNHANES VII-1. The middle-aged single-person household had lower subjective health status, more depression, and significantly lower quality of life than a multi-person household. Furthermore, middle-aged single-person households had lower KHEI scores compared to multi-person households. Middle-aged men living in single-person households had significantly lower KHEI scores, which included an adequate breakfast, mixed grain, and fruit intake, a component in the adequacy domain. Meanwhile, among middle-aged women living in single-person households, there was no significant difference in KHEI scores. In addition, men have a higher risk of depression in single-person households than in multi-person households. Among middle-aged single-person households in women, stress and depression increased in the lower KHEI group compared to the higher KHEI group (Figure 1).

According to this study, middle-aged single-person households for both sexes had lower household income levels and higher smoking rates, and lower levels of education for women than multi-person households. With increasing age, the non-voluntary single-person households occurred more frequently than households by voluntary choice [8]. There is a concern that the increase in the number of single households with middle-aged adults could lead to poverty-stricken single households of the elderly in the future [12].

Regarding studies on single-person households’ health-related quality of life, middle-aged single-person households have a low health-related quality of life as they demonstrate activity constraints, higher depression scores, do not practice physical activities, and have higher smoking rates. Senior citizens living alone have a low health-related quality of life as they have poor subjective health and activity constraints [37]. In a study of Chinese people [38], the health-related quality of life of low-income subjects was lower than high-income subjects. Song et al. stated that middle-aged people in single households had low quality of life compared to multi-person households [12]. For single-person households, a sense of social and economic deprivation is not reduced with time, so they are more vulnerable than multi-person households. Since they have a high sense of social and economic deprivation, they are more likely to be depressed [39]. Therefore, a social multidisciplinary approach is necessary to improve their vulnerable economic conditions that promote health-related quality of life. Specifically, eating alone was related to a greater likelihood of depression and suicidal ideation [40,41]. Similar to the research of Kang et al. that found that eating with family has a positive effect on mental health [42], this study also found that people in single-person households who ate meals alone were more likely to be depressed.

In this study, in women, protein intake of single-person households was significantly lower than that of multi-person households. Men who live alone also demonstrate lower KHEI scores and total scores in the frequency of breakfast, intake of mixed grains and fruits, and intake of sweets and beverages, among components of KHEI. Similar to these results, single-person households in their 40~50 s show lower interest in health than multi-person households, consume irregular meals, have low satisfaction levels with current eating patterns, and have a low alcohol drinking rate [9]. Single-person households show higher risks in delivery/takeaway meals and skipping three meals a day [43]. In the case of single-person households in their 19~29, they consume less dietary fiber and fluid than multi-person households, and as their education level and income level are higher, their cholesterol intake is higher [44].

Previous studies reported that living alone was associated with depression in older adults [45,46]. Hu et al. reported that older people living alone had a higher risk of depression than those who did not live alone (OR: 1.44; 95% CI: 1.04–1.99) in the qualitative meta-analysis [46]. Likewise, the results of this study show that the risk of depression increased in single-person household men by 6.262 times and single-person household women by 3.114 times compared with multi-person households. When socio-economic factors are adjusted, men in single-person households who have poor quality of meals have a 3.548-fold higher risk of depression than in multi-person households.

In this study, for women, the ORs for stress perception and depression with the T1 group of KHEI in single-person households compared to the T3 group of KHEI were 2.8 and 28.3, respectively. This is related to the frequency of breakfast, intake of mixed-grain and fruits and energy intake of sweets and beverages, which are low in KHEI items. The results of this study were similar to those reported by Yoon and Oh [29], who reported that the higher the KHEI index, the significantly reduced stress, depression, and suicidal ideation in women 19 years and older. A study by Lee and Lee [47] showed that women aged between 75 and 79 had a significantly improved subjective health status, quality of life and reduced risk of depression according to the increase of the KHEI index. Wang et al. [22] reported that a higher HEI-2015 score is associated with a lower risk of depression in adults from the National Health and Nutrition Examination Survey (2005–2016) in the USA.

Miki et al. [48] examined the association between dietary fiber and depressive symptoms in 1977 Japanese workers aged 19–69 years. They found that higher dietary fiber intake from vegetables and fruits had a significant effect on reducing depressive symptoms. McMartin et al. [49] reported the relation between fruit and vegetable intake and mental health disorders using a cross-sectional study of Canadians (*n*, 296, 121 aged 12 or older) five times between 2000 and 2009. Fruit and vegetable consumption has been proven to be negatively associated with depression, psychological distress, and poor mental health. Jacka et al. [50,51] investigated that a traditional Norwegian and Australian diet, which included vegetables, fruits, meat, fish, and whole grains, was also associated with lower rates of depression. Dietary recommendations to prevent depression include eating fruits, vegetables, legumes, whole grains, and nuts. [52]. In a recent meta-analysis of 21 studies from ten countries, a dietary pattern characterized by a high intake of fruit, vegetables, whole grain, fish, olive oil, low-fat dairy, and antioxidants, and low intake of animal foods was associated with a reduced risk of depression [53].

This study showed that middle-aged single-person households had lower subjective health status, higher levels of depression, and lower quality of life. Low diet quality was associated with increased stress and depression among middle-aged single-person women households. These results suggest a direction for diet management in single-person households with stress and depression. Based on the above results, the basis for the establishment of policies to improve the mental health of middle-aged single-person households in Korea is provided, and it is necessary to prepare an intervention plan with a multidisciplinary approach of constant interest to improve middle-aged single-person households’ quality of diet and health.

This study has several limitations. Since the KNHANES used in this study is a cross-sectional study, it is difficult to explain the causal relationship. Secondly, a limited number of variables were used for living alone. For example, the middle-aged who lived alone may have had poor social networks. Third, the evaluation of mental health status is self-reported data through questionnaires, and fourth, there may be possible researcher biases in the dietary intake survey. Fifth, the sample sizes in single-person households and multi-person households have a vast difference. However, the characteristics of single-person households shown in this study are similar to those reported in several studies [9,12,43]. Nevertheless, this study is meaningful in that it analyzed the characteristics of the healthy eating index, which can identify the important dietary factors for the mental health of single-person households. Also, this is the first reported study to analyze the relationship between a healthy eating index and mental health of middle-aged Korean single-person households.

## 5. Conclusions

This study determined the association between Korean Healthy Eating Index (KHEI) scores and mental health in middle-aged adults living in single-person households based on the 2016–2018 Korea National Health and Nutrition Examination Survey (KNHANES VII-1). In both sexes, single-person households had significantly lower subjective health status scores, lower EQ-5D index scores, and higher PHQ-9 index scores compared to multi-person households. Men had a higher risk of depression in single-person households than in multi-person households. Among middle-aged single-person households in women, stress and depression increased in the lower KHEI group compared to the higher KHEI group. Therefore, in order to improve the health-related quality of life of single-person middle-aged households, it is necessary to prepare an intervention plan with a multidisciplinary approach with continuous interest. In addition, it will be necessary to establish appropriate approaches and strategies for health management for middle-aged adults living in single-person households.

## Figures and Tables

**Figure 1 ijerph-19-04692-f001:**
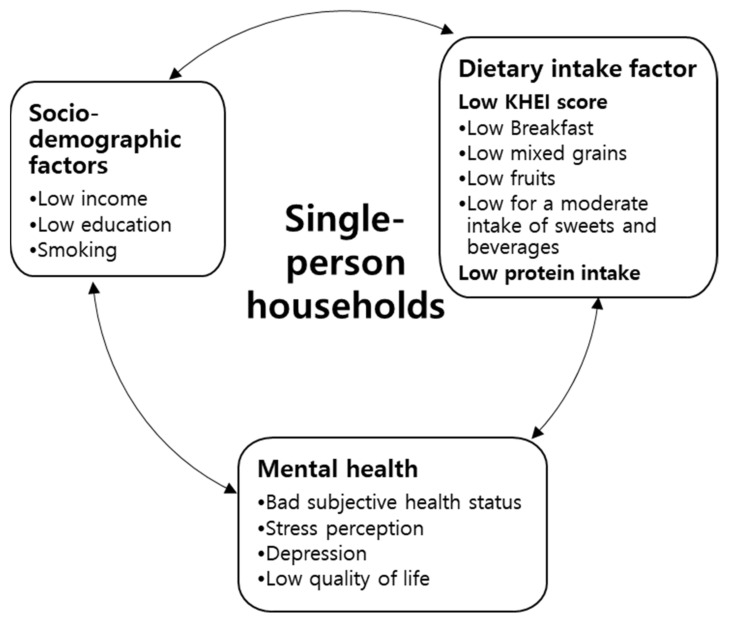
Key findings between dietary factor s and mental health in single-person households.

**Table 1 ijerph-19-04692-t001:** Socio-demographic characteristics by participants according to household size by sex.

Characteristics		Men (*n* = 1334)			Women (*n* = 2185)		
		Single-Person (*n* = 114)	Multi-Person (*n* = 1220)	*p*-Value ^3^	Single-Person (*n* = 144)	Multi-Person (*n* = 2041)	*p*-Value
Age (yrs)		49.19 ± 0.54 ^1^	49.59 ± 0.18	0.484	51.11 ± 0.61	49.65 ± 0.16	0.021
Education	≤Elementary	8 (6.22) ^2^	55 (4.96)	0.337	25 (15.50)	151 (7.05)	<0.001
	≤Middle school	15 (10.81)	99 (7.19)		20 (14.69)	191 (9.64)	
	≥High school	91 (82.97)	1066 (87.86)		99 (69.81)	1699 (83.31)	
Household income	Low	22 (19.74)	76 (6.79)	0.001	36 (26.53)	144 (7.46)	<0.0001
	Middle–low	26 (20.47)	243 (19.76)		50 (33.21)	422 (20.32)	
	Middle–high	24 (24.70)	388 (19.76)		28 (18.51)	655 (32.68)	
	High	42 (35.10)	511 (41.68)		30 (21.76)	819 (39.54)	
Marrital status	Single	90 (80.24)	104 (9.44)	<0.0001	106 (73.00)	212 (10.68)	<0.0001
	Married	24 (19.76)	1116 (90.56)		38 (27.00)	1829 (89.32)	
Alcohol intake	No	38 (31.43)	335 (28.22)	0.491	83 (57.51)	1096 (52.47)	0.304
	Yes	76 (68.57)	885 (71.78)		61 (42.49)	945 (47.54)	
Smoking status	No	45 (39.13)	748 (61.84)	<0.0001	125 (84.97)	1946 (94.81)	<0.0001
	Yes	69 (60.87)	472 (38.16)		19 (15.03)	95 (5.19)	
Physical activity	No	63 (55.67)	692 (55.16)	0.929	89 (58.03)	1156 (54.99)	0.518
	Yes	51 (44.33)	528 (44.84)		55 (41.97)	885 (45.01)	
Body mass index (kg/m^2^)		24.30 ± 0.39	24.76 ± 0.10	0.256	23.96 ± 0.33	23.50 ± 0.09	0.177
	Underweight (BMI < 18.5 kg/m^2^)	2 (2.41)	24 (2.20)	0.061	4 (2.93)	74 (3.44)	0.349
	Normal (18.5 ≤ BMI < 23 kg/m^2^)	32 (29.36)	324 (26.59)		59 (39.48)	950 (46.96)	
	Overweight (23.0 ≤ BMI < 25 kg/m^2^)	39 (36.93)	316 (25.85)		41 (27.09)	436 (21.47)	
	Obesity (BMI ≥ 25 kg/m^2^)	41 (31.30)	556 (45.36)		40 (30.50)	581 (28.13)	

^1^ Values are expressed as means ± standard error. ^2^ Values are expressed as numbers (%). ^3^ The *p*-value was estimated using the chi-square test and *t*-test in complex sample survey data analysis.

**Table 2 ijerph-19-04692-t002:** Participants’ mental health status according to household size by sex.

Variables	Men (*n* = 1334)			Women (*n* = 2185)		
	Single-Person (*n* = 114)	Multi-Person (*n* = 1220)	*p*-Value ^3^	Single-Person (*n* = 144)	Multi-Person (*n* = 2041)	*p*-Value
Subjective health status	2.98 ± 0.09 ^1^	3.23 ± 0.02	0.015	2.84 ± 0.08	3.15 ± 0.02	<0.001
Good	86 (75.02) ^2^	1052 (86.95)	<0.001	402 (69.95)	1693 (83.18)	<0.001
Poor	28 (24.98)	168 (13.05)		42 (30.05)	348 (16.82)	
Stress perception	2.16 ± 0.07	2.18 ± 0.02	0.813	2.26 ± 0.07	2.16 ± 0.02	0.160
No	85 (74.49)	896 (73.51)	0.845	99 (68.17)	1528 (75.59)	0.086
Yes	29 (25.51)	324 (26.49)		45 (31.83)	513 (24.41)	
PHQ-9 index	3.28 ± 0.52	1.70 ± 0.10	0.004	3.82 ± 0.43	2.29 ± 0.08	<0.001
Normal	102 (87.41)	1194 (97.75)	<0.0001	128 (89.44)	1963 (96.35)	<0.0001
Depression	12 (12.59)	26 (2.25)		16 (10.56)	78 (3.65)	
EQ-5D index	0.95 ± 0.01	0.97 ± 0.00	0.011	0.93 ± 0.01	0.96 ± 0.00	0.003

^1^ Values are expressed as means ± standard error. ^2^ Values are expressed as numbers (%). ^3^ The *p*-value was estimated using the chi-square test and *t*-test in complex sample survey data analysis.

**Table 3 ijerph-19-04692-t003:** Participants’ nutrient intake according to household size by sex.

Variables	Men (*n* = 1334)			Women (*n* = 2185)		
	Single-Person (*n* = 114)	Multi-Person (*n* = 1220)	*p*-Value ^2^	Single-Person (*n* = 144)	Multi-Person (*n* = 2041)	*p*-Value
Total energy (kcal)	2229.77 ± 85.43 ^1^	2310.44 ± 22.15	0.370	1728.32 ± 57.46	1672.62 ± 15.34	0.349
% energy of carbohydrate	63.68 ± 1.33	63.52 ± 0.35	0.909	65.78 ± 0.90	65.39 ± 0.28	0.682
% energy of protein	16.43 ± 0.64	15.74 ± 0.14	0.301	14.39 ± 0.35	14.78 ± 0.11	0.287
% energy of fat	19.89 ± 0.92	20.73 ± 0.29	0.387	19.84 ± 0.74	19.82 ± 0.23	0.985
Protein (g/1000 kcal)	36.45 ± 1.46	35.83 ± 0.30	0.683	34.39 ± 0.78	36.19 ± 0.27	0.031
Fat (g/1000 kcal)	19.84 ± 0.93	21.10 ± 0.29	0.196	21.16 ± 0.79	21.62 ± 0.26	0.583
Carbohydrate (g/1000 kcal)	143.52 ± 4.19	147.44 ± 1.12	0.376	160.02 ± 2.91	161.01 ± 0.74	0.742
Total dietary fiber (g/1000 kcal)	11.41 ± 0.56	12.35 ± 0.16	0.108	15.50 ± 0.60	15.36 ± 0.18	0.822
Calcium (mg/1000 kcal)	263.72 ± 16.78	263.81 ± 4.09	0.996	304.46 ± 13.69	294.96 ± 3.83	0.498
Phosphorus (mg/1000 kcal)	533.12 ± 17.23	537.19 ± 4.17	0.817	564.65 ± 12.94	575.83 ± 3.86	0.406
Iron (mg/1000 kcal)	5.99 ± 0.26	6.10 ± 0.08	0.673	6.58 ± 0.26	6.62 ± 0.07	0.889
Sodium (mg/1000 kcal)	1815.70 ± 72.32	1811.50 ± 23.61	0.956	1717.69 ± 66.31	1748.98 ± 21.89	0.646
Potassium (mg/1000 kcal)	1381.32 ± 46.01	1418.15 ± 16.32	0.430	1675.20 ± 55.36	1659.88 ± 14.44	0.785
Vitamin A(ugRAE/1000 kcal)	189.45 ± 17.20	190.00 ± 7.63	0.977	251.51 ± 18.61	213.86 ± 4.21	0.042
Thiamin (mg/1000 kcal)	0.70 ± 0.04	0.69 ± 0.01	0.900	0.70 ± 0.03	0.69 ± 0.01	0.903
Riboflavin (mg/1000 kcal)	0.76 ± 0.04	0.79 ± 0.01	0.377	0.88 ± 0.03	0.86 ± 0.01	0.524
Niacin (mg/1000 kcal)	6.54 ± 0.27	6.84 ± 0.08	0.285	6.89 ± 0.24	7.24 ± 0.07	0.157
Vitamin C (mg/1000 kcal)	26.89 ± 5.01	30.27 ± 1.11	0.510	37.58 ± 3.87	40.26 ± 1.18	0.511

^1^ Values are expressed as means ± standard error. ^2^ The *p*-value was estimated using the *t*-test in complex sample survey data analysis.

**Table 4 ijerph-19-04692-t004:** Korean Healthy Eating Index (KHEI) score of participants according to household size by sex.

Variables	Men (*n* = 1334)			Women (*n* = 2185)		
	Single-Person (*n* = 114)	Multi-Person (*n* = 1220)	*p*-Value ^2^	Single-Person (*n* = 144)	Multi-Person (*n* = 2041)	*p*-Value
Total KHEI score (0–100)	57.74 ± 1.28 ^1^	63.68 ± 0.43	<0.0001	65.99 ± 1.25	66.53 ± 0.37	0.681
Component of KHEI score						
Adequacy						
Breakfast (0–10)	5.52 ± 0.47	7.55 ± 0.13	<0.0001	6.86 ± 0.38	7.51 ± 0.10	0.100
Mixed grains (0–5)	0.93 ± 0.18	2.06 ± 0.07	<0.0001	1.97 ± 0.21	2.11 ± 0.05	0.519
Total fruits (0–5)	1.28 ± 0.20	1.91 ± 0.07	0.003	2.85 ± 0.21	2.88 ± 0.06	0.868
Fresh fruits (0–5)	1.28 ± 0.21	2.21 ± 0.07	<0.0001	3.06 ± 0.21	3.12 ± 0.07	0.765
Total vegetable (0–5)	3.94 ± 0.14	3.91 ± 0.04	0.816	3.47 ± 0.14	3.38 ± 0.03	0.565
Vegetable, excluding kimchi and pickles (0–5)	3.53 ± 0.15	3.44 ± 0.05	0.551	3.31 ± 0.14	3.14 ± 0.04	0.235
Meat, fish, eggs, and legumes (0–10)	7.09 ± 0.35	7.42 ± 0.09	0.373	6.86 ± 0.30	6.86 ± 0.09	0.986
Milk and dairy (0–10)	2.95 ± 0.44	3.03 ± 0.14	0.873	3.83 ± 0.40	3.41 ± 0.12	0.298
Moderation						
Sodium (0–10)	5.77 ± 0.32	5.52 ± 0.11	0.457	7.44 ± 0.29	7.71 ± 0.07	0.356
Saturated fatty acid (0–10)	7.34 ± 0.41	7.55 ± 0.12	0.629	7.80 ± 0.30	7.69 ± 0.10	0.738
Sweets and beverages (0–10)	8.67 ± 0.29	9.26 ± 0.07	0.049	9.28 ± 0.18	9.28 ± 0.05	0.999
Balance						
Carbohydrate (0–5)	2.55 ± 0.21	2.83 ± 0.07	0.222	2.70 ± 0.18	2.66 ± 0.06	0.796
Fat (0–5)	3.61 ± 0.17	3.69 ± 0.06	0.653	3.49 ± 0.18	3.55 ± 0.05	0.778
Total energy (0–5)	3.28 ± 0.22	3.31 ± 0.07	0.900	3.07 ± 0.20	3.22 ± 0.05	0.440

^1^ Values are expressed as means ± standard error. ^2^ The *p*-value was estimated using the *t*-test in complex sample survey data analysis.

**Table 5 ijerph-19-04692-t005:** ORs for mental health-related outcomes according to household size by sex.

Variables			Men (*n* = 1334)			Women (*n* = 2185)	
		Single-Person (*n* = 114)	Multi-Person (*n* = 1220)	*p*-Value	Single-Person (*n* = 144)	Multi-Person (*n* = 2041)	*p*-Value
Subjective health status (poor)	Model 1 ^2^	2.2 (1.4–3.6) ^1^	1.0 (ref.)	0.001	2.1 (1.4–3.2)	1.0 (ref.)	<0.001
	Model 2 ^3^	1.4 (0.8–2.5)	1.0 (ref.)	0.250	1.5 (0.9–2.4)	1.0 (ref.)	0.137
Stress perception (yes)	Model 1	1.0 (0.6–1.6)	1.0 (ref.)	0.847	1.4 (0.9–2.2)	1.0 (ref.)	0.088
	Model 2	1.0 (0.5–1.8)	1.0 (ref.)	0.905	1.1 (0.7–1.7)	1.0 (ref.)	0.816
Depression	Model 1	6.3 (2.7–14.3)	1.0 (ref.)	<0.0001	3.1 (1.7–5.6)	1.0 (ref.)	<0.001
	Model 2	3.5 (1.2–10.1)	1.0 (ref.)	0.018	1.1 (0.6–2.0)	1.0 (ref.)	0.852

^1^ Values are expressed as odds ratios (95% confidence intervals). ^2^ Model 1: crude. ^3^ Model 2: adjusted for age, marital status, household income, and smoking in men; adjusted for age, education, marital status, household income, and smoking in women.

**Table 6 ijerph-19-04692-t006:** ORs for mental health-related outcomes according to KHEI score by household size in men.

Variables			Single-Person (*n* 114)				Multi-Person (*n* = 1220)		
			Tertile of KHEI				Tertile of KHEI		
		T1 (*n* = 38)	T2 (*n* = 38)	T3 (*n* = 38)	*p* for trend	T1 (*n* = 406)	T2 (*n* = 407)	T3 (*n* = 407)	*p* for Trend
Subjective health status (poor)	Model 1 ^2^	1.4 (0.5–4.4) ^1^	0.8 (0.2–2.9)	1.0 (ref.)	0.638	1.5 (0.9–2.4)	1.2 (0.8–2.0)	1.0 (ref.)	0.084
	Model 2 ^3^	1.3 (0.4–4.5)	0.5 (0.1–2.1)	1.0 (ref.)	0.860	1.4 (0.9–2.2)	1.2 (0.8–1.9)	1.0 (ref.)	0.183
Stress perception (yes)	Model 1	1.3 (0.4–4.5)	1.1 (0.3–4.1)	1.0 (ref.)	0.697	1.3 (1.0–1.9)	1.1 (0.8–1.5)	1.0 (ref.)	0.084
	Model 2	1.2 (0.3–4.1)	1.2 (0.3–4.7)	1.0 (ref.)	0.775	1.2 (0.9–1.7)	1.1 (0.8–1.4)	1.0 (ref.)	0.263
Depression	Model 1	3.1 (0.6–17.0)	0.9 (0.1–8.1)	1.0 (ref.)	0.255	2.7 (0.9–8.0)	1.4 (0.4–4.4)	1.0 (ref.)	0.055
	Model 2	2.3 (0.4–14.2)	0.7 (0.1–5.1)	1.0 (ref.)	0.418	1.8 (0.7–5.0)	1.2 (0.4–3.8)	1.0 (ref.)	0.214

^1^ Values are expressed as odds ratios (95% confidence intervals). ^2^ Model 1: crude. ^3^ Model 2: adjusted for age, marital status, household income, and smoking.

**Table 7 ijerph-19-04692-t007:** ORs for mental health-related outcomes according to KHEI score by household size in women.

Variables			Single-Person (*n* = 144)				Multi-Person (*n* = 2041)		
			Tertile of KHEI				Tertile of KHEI		
		T1 (*n* = 48)	T2 (*n* = 48)	T3 (*n* = 48)	*p* for Trend	T1 (*n* = 680)	T2 (*n* = 681)	T3 (*n* = 680)	*p* for Trend
Subjective health status (poor)	Model 1 ^2^	1.5 (0.6–3.6) ^1^	1.0 (0.4–2.7)	1.0 (ref.)	0.385	1.3 (0.9–1.8)	0.9 (0.7–1.3)	1.0 (ref.)	0.192
	Model 2 ^3^	1.2 (0.5–3.1)	0.8 (0.3–2.4)	1.0 (ref.)	0.764	1.2 (0.8–1.7)	0.9 (0.6–1.3)	1.0 (ref.)	0.340
Stress perception (yes)	Model 1	3.5 (1.3–9.0) *	1.1 (0.4–3.1)	1.0 (ref.)	0.013	1.4 (1.0–1.8) *	1.0 (0.7–1.3)	1.0 (ref.)	0.035
	Model 2	2.8 (1.0–7.5) *	1.1 (0.4–3.3)	1.0 (ref.)	0.051	1.3 (0.9–1.6)	0.9 (0.7–1.3)	1.0 (ref.)	0.182
Depression	Model 1	25.4 (3.2–203.8) **	7.2 (0.7–79.5)	1.0 (ref.)	0.002	2.2 (1.1–4.6) *	1.7 (0.8–3.6)	1.0 (ref.)	0.027
	Model 2	28.3 (2.1–374.2) *	5.2 (0.4–61.4)	1.0 (ref.)	0.007	1.9 (0.9–4.1)	1.5 (0.7–3.4)	1.0 (ref.)	0.092

^1^ Values are expressed as odds ratios (95% confidence intervals). ^2^ Model 1: crude. ^3^ Model 2: adjusted for age, education, marital status, household income, and smoking. * *p* < 0.05, ** *p* < 0.01.

## Data Availability

The KNHANES data used in the manuscript can be found at the following link: https://knhanes.kdca.go.kr/knhanes/main.do (accessed on 21 December 2021).

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
