# Peer review of "Association between Healthy Eating Index and Mental Health in Middle-Aged Adults Based on Household Size in Korea"

_ijerph, 2022, doi:10.3390/ijerph19084692_

Round 1

Reviewer 1 Report

Thank you for giving me the opportunity to review the manuscript entitled “Association between Healthy Eating Index and Mental Health in Middle-Aged Adults based on Household Size in Korea”, which explores the association between dietary lifestyle and mental health outcomes among middle-aged individuals living alone. The objective of this study is of great interest and adds significant information to previous knowledge. The design is appropriate and the results and findings are reliable. The manuscript needs only minor revisions:

  1. Abstract, lines 16-17: I would suggest writing out EQ-5D and PHQ-9 for better readability.
  2. Abstract, lines 20-21: please explain, what are T1 and T3 groups?
  3. Abstract: for the ORs, I would suggest using only one decimal point.
  4. Materials and Methods, line 164: Did you mean sugar-sweetened beverages or soft drinks?
  5. Results, lines 205-212: What do you mean by soft drinks group?
  6. Table 1: I think, below Body mass index (kg/m2), the term obesity is obsolete. And in the column of Men & Multi-person, the numbers of overweight and obesity (please use obesity instead of obese) are missing.
  7. Whole manuscript: Could you please check, if you used people’s first language regarding obesity? Please use persons/participants with obesity and not obese.
  8. Results, line 220: I think it should mean showed instead of were.
  9. Table 4, line milk and dairy: There might be a tipping error: 0.8733.83
  10. Results, lines 256-281 and Table 5, 6 & 7: I would suggest using only one decimal point when showing ORs. Please, also check it in the discussion
  11. Results, lines 270-271: Could you please specify what T1 and T3 groups are? It is clear when I have a look at Table 7 but not in the text alone.
  12. Discussion, line 391: Tipping error: middles-aged
  13. Conclusion, line 405: I would rather use the past tense.

Author Response

  1. Abstract, lines 16-17: I would suggest writing out EQ-5D and PHQ-9 for better readability.

Response 1. We wrote out as EuroQol-5D and Patient Health Questionnaire-9. Line 17.

  1. Abstract, lines 20-21: please explain, what are T1 and T3 groups?

Response 2. We explained T1 as the first tertile group of KHEI scores and T3 group as the third tertile group. Line 20-21.

  1. Abstract: for the ORs, I would suggest using only one decimal point.

Response 3. We used only one decimal point.

  1. Materials and Methods, line 164: Did you mean sugar-sweetened beverages or soft drinks?

Response 4. We changed “sweets and beverages, and sodium” Line 174.

Sweets and beverages included sugars, confectionary, coffee or tea, cocoa, alcoholic beverages, soft drinks, fruit and vegetable drinks, and other beverages. Line 176-177.

  1. Results, lines 205-212: What do you mean by soft drinks group?

Response 5. We deleted that wrong paragraph.

  1. Table 1: I think, below Body mass index (kg/m2), the term obesity is obsolete. And in the column of Men & Multi-person, the numbers of overweight and obesity (please use obesity instead of obese) are missing.

Response 6. We have corrected the points pointed out in Table 1.

  1. Whole manuscript: Could you please check, if you used people’s first language regarding obesity? Please use persons/participants with obesity and not obese.

Response 7. We checked there were no errors.

  1. Results, line 220: I think it should mean showed instead of were.

Response 8. We changed as “showed”. Line 224.

  1. Table 4, line milk and dairy: There might be a tipping error: 0.8733.83

Response 9. We corrected error.

  1. Results, lines 256-281 and Table 5, 6 & 7: I would suggest using only one decimal point when showing ORs. Please, also check it in the discussion

Response 10. We used only one decimal point.

  1. Results, lines 270-271: Could you please specify what T1 and T3 groups are? It is clear when I have a look at Table 7 but not in the text alone.

Response 11. We specified T1 group as the first tertile (T1) group and T3 group as the third tertile (T3) group. Line 274-275.

  1. Discussion, line 391: Tipping error: middles-aged

Response 12. We changed as ‘middle-aged’. Line 402

  1. Conclusion, line 405: I would rather use the past tense.

Response 13. We changed as ‘Men had’. Line 420.

Reviewer 2 Report

Lines 16,17

Please provide the meaning of EQ-5D and PHQ-9, since this is the first time to appear in text

Line 33

(… developed)

Do you mean considered ?

Line 60

( … goose fathers)

Please explain the term

Line 140

(… EuroQol-5D)

Please explain further this tool (eg health-related questionnaire etc)

Lines 195-198

Please add years after numbers

Line 206

(…FFO method)

Please explain the term (eg food frequency questionnaires)

Line 314

Please give more information on the lower income levels

Additionally

The construction of a specific figure, summarizing all key findings between single and multi person households, could be very useful to the reader

Author Response

Comments and Suggestions for Authors

  1. Lines 16,17, Please provide the meaning of EQ-5D and PHQ-9, since this is the first time to appear in text

 Response 1. We changed as EuroQol-5D and Patient Health Questionnaire-9. Line 17.

  1. Line 33 (… developed) Do you mean considered ?

Response 2. We used ‘considered’ instead of ‘developed’. Line 34.

  1. Line 60 ( … goose fathers) Please explain the term

Response 3. We changed the sentences as this.

The increase in middle-aged single-person households is attributed to singles, divorcees, or goose fathers, who work alone in Korea to pay for their families that live abroad or in a metropolitan area for educational reasons for their children [9, 10, 11]. Line 48-50

  1. Line 140 (… EuroQol-5D)

Please explain further this tool (eg health-related questionnaire etc)

Response 4. We explained further this tool. Line 147-148.

Health-related quality of life was measured by EuroQol-5D (EQ-5D) questionnaire of the health survey, which was developed by the EuroQoL Group .

  1. Lines 195-198 Please add years after numbers

Response 5. We added years.

  1. Line 206 (…FFO method)

Please explain the term (eg food frequency questionnaires)

Response 6. We deleted that wrong paragraph.

  1. Line 314: Please give more information on the lower income levels

Response 7. We rephrased that sentence more clearly. Household incomes were categorized as low, middle-low, middle-high, and high, such as mentioned in method.

Line 322-324.

According to this study, middle-aged single-person households for both sexes had lower household income levels and higher smoking rates, and lower levels of education for women than multi-person households.

  1. Additionally, the construction of a specific figure, summarizing all key findings between single and multi person households, could be very useful to the reader.

Response 8. We added a figure summarized key findings. Line 305-307.

Reviewer 3 Report

Brief summary  Thank you for the opportunity to review your paper! You will see many comments that are offered to help to strengthen the presentation of your study.   This paper is well structured and is an important addition to the literature. The study aimed to determine the relationship between the Korean Healthy Eating Index (KHEI) scores and mental health in middle-aged adults living in single-person homes, using data from the Korea National Health and Nutrition Examination Survey. The main strengths of this paper are that it addresses an interesting and timely question and that the scientific methods and assumptions are valid and clearly outlined. Furthermore, the description of data and the calculations are sufficiently complete to be followed and allow their reproduction by fellow scientists.   General concept comments This is a fascinating article and an important area of investigation. However, this paper has some shortcomings related to its depth in the introduction and discussion. Additionally, the manuscript could benefit from moderate technical editing to correct grammatical errors, wordiness (e.g., condensing/combining sentence), and improve clarity (some edits suggested in the review, but not all). This manuscript could provide much-needed research to the field; however, the wordiness distracts from the manuscript's intended impact as it is currently written.   Main suggestions for improvement:

  1. Introduction: Incorporating additional citations to help place the current work in a better context of the scientific progress and provide more accurate information on this topic. Furthermore, the literature review could be slightly more critical to highlight the importance and need for this study.  
  2. Technical editing: Improving the conciseness and flow of the paper will strengthen its impact. Currently, there are a few paragraphs that include nearly identical sentences; for example, lines 59-63 state, “The increase in middle-aged single-person households is attributed to singles, divorcees, or goose fathers, who stay away from their families for their children’s education. The increase in the number of single households may also be attributed to the phenomenon of goose fathers, who work alone in Korea to pay for their families that live abroad or in a metropolitan area for educational reasons for their children [16, 17, 18].” These sentences could be combined to increase succinctness. Additional suggestions include: adding transition phrases between sentences, transition sentences to link paragraphs, and condensing and combining sentences.
  3. Discussion and Conclusion: Expand upon and include the study limitations, research implications, and directions for future research in the discussion more specifically.

Overall, this is an important study and should be considered for publication in IJERPH once these concerns have been addressed.     Specific comments  Most comments below are not criticisms to be addressed absolutely, but mere suggestions for improvement.     Abstract No comments to add   Introduction Overall: Research questions and goals are thoroughly summarized, and the introduction is completed with a clear statement of purpose for the current study. I have suggested some areas where the authors should consider combining sentences to allow the text to flow more smoothly. Additionally, the study states that few studies have investigated the phenomenon of interest; however, I quickly reviewed the literature regarding the HEI and mental health and found that the literature is more extensive than it is currently presented in the paper.  I have provided some examples below that will help give the reader what they may need to know to understand this study's context. Overall, the introduction is clear and logical; it just needs to be slightly reworked to adequately summarize the existing body of knowledge related to the healthy eating index and mental health.    Lines 32-33: Consider adding a transition sentence to improve the flow of your text.    Line 33: remove the comma before developed   Lines 33-34: A concluding sentence to tie paragraphs one and two together would help increase the flow of the text   Line 34: This sentence is a little unclear. Rephrasing this sentence may help with the sentence’s clarity.   Line 38: This sentence is a little unclear. Rephrasing this sentence may help with the sentence’s clarity.   Lines 39-41: This sentence should be rephrased to explain why these facts are essential.    Line 42: Change *visit* to visits   Line 44: Does non-single refer to those who are only married? Or does it also include living with friends or family?    Lines 59-63: Sentences should be combined and incorporated into the 3rd paragraph.    Lines 64-71: Extensive qualitative literature examining the relationship between communal eating practices (food preparation, food sharing, and food consumption) and well-being could further strengthen the literature review.    Lines 64-67: Combine these sentences as nearly identical information.  Below are articles regarding:     (1) Studies regarding Youth and adolescents: (Systematic Review) O’neil, A., Quirk, S. E., Housden, S., Brennan, S. L., Williams, L. J., Pasco, J. A., ... & Jacka, F. N. (2014). Relationship between diet and mental health in children and adolescents: a systematic review. American journal of public health, 104(10), e31-e42.   Jacka, F. N., Kremer, P. J., Berk, M., de Silva-Sanigorski, A. M., Moodie, M., Leslie, E. R., ... & Swinburn, B. A. (2011). A prospective study of diet quality and mental health in adolescents. PloS one, 6(9), e24805.   Jacka, F. N., Rothon, C., Taylor, S., Berk, M., & Stansfeld, S. A. (2013). Diet quality and mental health problems in adolescents from East London: a prospective study. Social psychiatry and psychiatric epidemiology, 48(8), 1297-1306.   Kulkarni, A. A., Swinburn, B. A., & Utter, J. (2015). Associations between diet quality and mental health in socially disadvantaged New Zealand adolescents. European Journal of Clinical Nutrition, 69(1), 79-83.   McMartin, S. E., Kuhle, S., Colman, I., Kirk, S. F., & Veugelers, P. J. (2012). Diet quality and mental health in subsequent years among Canadian youth. Public health nutrition, 15(12), 2253-2258.   Dimov, S., Mundy, L. K., Bayer, J. K., Jacka, F. N., Canterford, L., & Patton, G. C. (2021). Diet quality and mental health problems in late childhood. Nutritional neuroscience, 24(1), 62-70.   Tomlinson, D., Wilkinson, H., & Wilkinson, P. (2009). Diet and mental health in children. Child and Adolescent Mental Health, 14(3), 148-155.   Rossa-Roccor, V., Richardson, C. G., Murphy, R. A., & Gadermann, A. M. (2021). The association between diet and mental health and wellbeing in young adults within a biopsychosocial framework. PloS one, 16(6), e0252358   (2) Healthy Eating Index and Mental Health and Wellbeing Kuczmarski, M. F., Sees, A. C., Hotchkiss, L., Cotugna, N., Evans, M. K., & Zonderman, A. B. (2010). Higher Healthy Eating Index-2005 scores associated with reduced symptoms of depression in an urban population: findings from the Healthy Aging in Neighborhoods of Diversity Across the Life Span (HANDLS) study. Journal of the American dietetic association, 110(3), 383-389.   Rahmani, J., Milajerdi, A., & Dorosty-Motlagh, A. (2018). Association of the Alternative Healthy Eating Index (AHEI-2010) with depression, stress and anxiety among Iranian military personnel. BMJ military health, 164(2), 87-91.   Exebio, J. C., Zarini, G. G., Exebio, C., & Huffman, F. G. (2011). Healthy Eating Index scores associated with symptoms of depression in Cuban-Americans with and without type 2 diabetes: a cross sectional study. Nutrition journal, 10(1), 1-7.   Saneei, P., Hajishafiee, M., Keshteli, A. H., Afshar, H., Esmaillzadeh, A., & Adibi, P. (2016). Adherence to Alternative Healthy Eating Index in relation to depression and anxiety in Iranian adults. British Journal of Nutrition, 116(2), 335-342.     (3)KHEI and Mental Health Yoon, Y. S., & Oh, S. W. (2021). Relationship between psychological distress and the adherence to the Korean healthy eating index (KHEI): the Korea National Health and Nutrition Examination Survey (KNHANES) 2013 and 2015. Nutrition Research and Practice, 15(4), 516-527.   Methods Overall: The methods, results, and discussion sections were all sufficient and explicit.    Lines 89 – 92: This sentence is a little unclear. Rephrasing this sentence may help with the sentence’s clarity.   Line 147: The authors should mention the limitations to 24 hours dietary recall.    Results Overall: The results are clearly and completely described in an organized manner.   Line 202: The women p-value is missing   Discussion and Conclusion Overall: The study results are well summarized and linked to the literature identified in the introduction section; though, as mentioned previously, I would suggest incorporating additional literature to allow the reader to understand the literature on this topic more strongly. Additionally, while the results are well summarized, I suggest that the authors draw additional inferences from the results—the limitations discussed need to be expanded and included in the discussion more concretely. For example, the sample size in single households (SH) vs. multi-person households (MPH) is vastly different. However, since many comparisons are made between SHs and MPHs, the authors should justify these populations' similarities. Furthermore, adding the evaluation and in-depth interpretation of key findings and the study implications concerning the original goals and hypotheses, in addition to the proposal of new directions for future research, will further enhance the paper’s conclusion section.    Line 323: add *a* before *low* (…have a low..)   Line 326: remove *that of*   Line 334: Add *the* before *research*    Line 337 – 338: This sentence is a little unclear. Rephrasing this sentence may help with the sentence’s clarity.   Line 343: Add *have* before *a low alcohol drinking rate*   Lines 356-358: This sentence is a little unclear. Rephrasing this sentence may help with the sentence’s clarity.   Lines 360 – 362: This sentence is a little unclear. Rephrasing this sentence may help with the sentence’s clarity.   Line 378: Include *the* before *consumption*   Line 381: Delete *apparently*   Line 385: Add *the* before *mental health*   Line 387: add *a* before *multidisciplinary*    Lines 389 – 398: Additional limitations include: self-reported data, possible researcher biases, etc.   Line 391: *middles-aged* should be *middle-aged*   Line 393: add *a* before *healthy eating index*          

Author Response

  1. Lines 32-33: Consider adding a transition sentence to improve the flow of your text.   

We added a transition sentence. Line 34-36.

People's eating habits have become very diverse, and it is necessary to evaluate the overall quality of eating habits using these dietary evaluation indexes.

  1. Line 33: remove the comma before developed

Response 2. We used ‘considered’ instead of ‘developed’. And removed the comma. (Line34)

  1. Lines 33-34: A concluding sentence to tie paragraphs one and two together would help increase the flow of the text  

Response 3. We added a transition sentence. Line 36-38.

Additionally, meals in single-person households often lack variety and specific core foods, such as fruits, vegetables, and fish, and are associated with unhealthy dietary patterns [5].

  1. Line 34: This sentence is a little unclear. Rephrasing this sentence may help with the sentence’s clarity.  

Response 4. We rephrased the sentence as this. Line 39-40.

The proportion of single-person households worldwide is increasing due to various reasons such as independence from parents, school, work, late marriage, and divorce.

  1. Line 38: This sentence is a little unclear. Rephrasing this sentence may help with the sentence’s clarity.  

Response 5. We rephrased the sentence as this. Line 43-44.

Single-person households are more exposed to risks such as income, housing, safety, and health compared to multi-person households.  

  1. Lines 39-41: This sentence should be rephrased to explain why these facts are essential.   

Response 6. We rephrased the sentence as this. Line 44-47.

In the case of single-person households, due to social isolation, depression, loneliness, and sadness are greater than those of multi-person households, and they are not active in participating in cultural activities [8].

  1. Line 42: Change *visit* to visits

Response 7. We changed it. Line 51.

  1. Line 44: Does non-single refer to those who are only married? Or does it also include living with friends or family?   

Response 8. Non-single households include married, living with friends or family.   

  1. Lines 59-63: Sentences should be combined and incorporated into the 3rd paragraph.   

Response 9. We combined and incorporated into 3rd paragraph. Line 48-50.

The increase in middle-aged single-person households is attributed to singles, divorcees, or goose fathers, who work alone in Korea to pay for their families that live abroad or in a metropolitan area for educational reasons for their children [9, 10, 11].

10.Lines 64-71: Extensive qualitative literature examining the relationship between communal eating practices (food preparation, food sharing, and food consumption) and well-being could further strengthen the literature review.   

Response 10. We added this sentence. Line 67-69.

A systematic review by Tamminen et al. [19] reported that a potential association between living alone and low positive mental health was found in three out of the four studies.”

We changed line 64-71 into new sentences. Line 69-76. We cited three references [20, 25, 28] recommended by reviewers. Thank you for your advice.

According to previous studies related to diet and mental health, many studies have been conducted on food and nutrition intake status and diet quality among adolescents [20], adults [21] and the elderly [22, 23]. A systematic review by GÅ‚Ä…bska et al. [24] reported that a high total intake of fruits and vegetables might protect against depressive symptoms. In addition, many studies have reported the relationship between dietary evaluation indices such as HEI [25-27] and alternative HEI [28] and mental health. Recently, Yoon and Oh [29] reported the relationship between KHEI and psychological distress in Korean adults.

  1. Lines 89 – 92: This sentence is a little unclear. Rephrasing this sentence may help with the sentence’s clarity.  

Response 11. We have rephrased the sentences clearly. Line 95-99.

Among the 6,198 who were aged 40-60 who participated in all three surveys (health and behavior interviews; health examinations; nutrition surveys), participants with abnormal dietary intake (men with < 800 kcal/day or > 4000 kcal/day and women with < 500 kcal/day or > 3500 kcal/day) (n = 278) were excluded [31].

  1. Line 147: The authors should mention the limitations to 24 hours dietary recall.    

Response 12. We have mentioned the limitations to 24 hours dietary recall. Line 158-160.

However, there are some disadvantages of this method; it relies on the participant's memory and is not able to describe the typical diet with a single day's intake.

13.Results Overall: The results are clearly and completely described in an organized manner.  

Line 202: The women p-value is missing   

Response 13. We corrected it. Line 213-214

  1. Discussion Line 323: add *a* before *low* (…have a low..)  

Response 14. We changed a low. Line 332.

  1. Line 326: remove *that of*  

Response 15. We removed ‘that of’. Line 335.

  1. Line 334: Add *the* before *research*   

Response 16. We added ‘the’ before ‘research’. Line 343.

  1. Line 337 – 338: This sentence is a little unclear. Rephrasing this sentence may help with the sentence’s clarity.  

Response 17. We rephrased the sentence as this. Line 346-347.

In this study, in women protein intake of single-person households was significantly lower than that of multi-person households.

  1. Line 343: Add *have* before *a low alcohol drinking rate*  

Response 18. We added ‘add’ before ‘a low alcohol drinking rate’. Line 352.

  1. Lines 356-358: This sentence is a little unclear. Rephrasing this sentence may help with the sentence’s clarity.  

Response 10. We rephrased the sentence as this. Line 365-367.

In this study, for women the ORs for stress perception and depression with T1 group of KHEI in single-person households compared to the T3 group of KHEI were 2.8 and 28.3, respectively.

  1. Lines 360 – 362: This sentence is a little unclear. Rephrasing this sentence may help with the sentence’s clarity.  

Response 20. We rephrased the sentence as this. Line 368-371.

The results of this study were similar to those reported by Yoon and Oh [25], who reported that the higher the KHEI index, the significantly reduced stress, depression, and suicidal ideation in women 19 years and older.

  1. Line 378: Include *the* before *consumption*  

Response 21. We included ‘the’ before ‘consumption’. Line 387.

  1. Line 381: Delete *apparently*  

Response 22. We deleted ‘apparently’. Line 390.

  1. Line 385: Add *the* before *mental health*  

Response 23. We added ‘the’ before ‘mental health’.  Line 396.

  1. Line 387: add *a* before *multidisciplinary*   

Response 24. We added ‘a’ before ‘multidisciplinary’. Line 398.

  1. in-depth analysis

Response 25. We added this sentence Line 392-395.

Low diet quality was associated with increased stress and depression among middle-aged single-person women households. These results suggest a direction for diet management in single-person households with stress and depression.

  1. Lines 389 – 398: Additional limitations include: self-reported data, possible researcher biases, etc.  

Response 26. We added additional limitations. Line 403-412.

Third, the evaluation of mental health status is self-reported data through questionnaires, and fourth, there may be possible researcher biases in the dietary intake survey. Fifth, the sample sizes in single-person households and multi-person households have vast difference. However, the characteristics of single-person households shown in this study are similar to those reported in several studies [9, 12, 45]. Nevertheless, this study is meaningful in that it analyzed the characteristics of the healthy eating index, which can identify the important dietary factors for the mental health of single-person households. Also, this is the first reported study to analyze the relationship between a healthy eating index and mental health of middle-aged Korean single-person households.

  1. Line 391: *middles-aged* should be *middle-aged*  

Response 27. We changed as ‘middle-aged’. Line 402.

  1. Line 393: add *a* before *healthy eating index*          

Response 28.  We added ‘a’ before ‘healthy eating index’. Line 411.

Round 2

Reviewer 3 Report

Lines 76 – 78: This sentence is repeated twice.

Below I have broken down the paragraphs of the introduction to easily identify its progression in supporting the authors’ research. The authors have a decent foundation for their introduction, unfortunately, it is not structured in a way to tell a story and convince the reader of their study’s importance. The introduction should be a logical flow of ideas that leads up to the hypothesis. I recommend that the authors pull apart their introduction and decide which ideas make sense to present first, second, third, etc. and think about how to transition between ideas. The hypotheses should flow logically out of everything that has already been presented, so that the hypothesis makes sense to the reader. Currently, the introduction includes wonderful ideas, but are not grouped together in a way that enables the reader to understand and appreciate the authors’ objectives.

OUTLINE:

First Paragraph:

  • Korean Healthy Eating Index (KHEI) considers the characteristics of Korean meals and national dietary guidelines.
    • Developed to monitor and evaluate the eating habits of a population group through data such as the National Health and Nutrition Survey, rather than assessing individual eating habits.
  • People's eating habits have become very diverse, and it is necessary to evaluate the overall quality of eating habits using these dietary evaluation indexes.
  • Additionally, meals in single-person households often lack variety and specific core foods, such as fruits, vegetables, and fish, and are associated with unhealthy dietary patterns.

Second Paragraph:

  • The proportion of single-person households worldwide is increasing
  • The Korean population is increasing
  • Single person households are exposed to more risks
  • Single households typical have increased mental health conditions
  • Single person households are increasing due to singles, divorcees, or goose fathers,
  • Health care visits and chronic diseases are higher in single person households
  • Living alone has higher negative health consequences

Third Paragraph

  • Quality of life description
  • Association between living alone and low positive mental health
  • Food and nutrition intake status and diet quality
  • High total intake of fruits and vegetables might protect against depressive symptoms
  • Relationship between dietary evaluation indices such as HEI and alternative HEI and mental health.
  • relationship between KHEI and psychological distress in Korean adults.

Fourth Paragraph

  • Study aimed to identify the association between Korean Healthy Eating Index (KHEI) scores and mental health in middle-aged adults living in single-person households, based on the 2016–2018 Korea National Health and Nutrition Examination Survey
  • The overall goal of the study is to guide those with stress and depression toward healthy eating behaviors by providing information on basic dietary data.
  •  

Author Response

April 7, 2022

Managing editor, IJERPH

Dear Fiona Han,

We are sending you the revised manuscript. Manuscript ID: ijerph-1626207

Title: Association between Healthy Eating Index and Mental Health in Middle-Aged Adults based on Household Size in Korea

1.We deleted the repeated sentence.

However, there are few studies on the relationship between KHEI and mental health.

  1. According to the reviewer's opinion, we grouped it into 4 paragraphs, moving the sentences to health in the second paragraph and mental health in the third paragraph (line 27-64). At the reviewer's request for logical flow, we moved the sentences to connect the whole flow and added conjunctions. We rearranged the number of references (numbers 1-18).

1) We made it into one sentence and presented it in the first sentence. (line 28-29)

2) We moved the position after editing the sentence. (line 39-41)

3) We have condensed two sentences into one. (line 50-53)

4) We moved the sentence to the third paragraph. (line 60-64)

5) We moved to the second paragraph. (line 53-55)

  1. We commissioned Editage and received English proofreading. We send the English proofreading certificate together.

Sincerely,

EunJung Lee, Ph.D

Associate professor, Food & Nutrition Major

* Corresponding author: EunJung Lee, Food and Nutrition Major,

School of Wellness Industry Convergence, Hankyong National University,

327 Jungang-no, Anseong-si, Gyonggi-do, 17579, Korea  

Tel : 82-31-670-5185 Fax : 82-31-670-5189 E-mail : [email protected]
